# Hydrodynamic Model of Diver–DPV Coupled Multi-Body and Its Underwater Cruising Numerical Simulation

**Hansheng Li, Fenglei Han \*, Haitao Zhu, Jiawei Zhang, Weipeng Zhang and Yuliang Wu**

Shipbuilding Engineering College, Harbin Engineering University, Harbin 150001, China; cnzwzero@outlook.com (H.L.); zhuhaitao@hrbeu.edu.cn (H.Z.); zhangjiawei@hrbeu.edu.cn (J.Z.); zhangweipeng@hrbeu.edu.cn (W.Z.); wuyuliang@hrbeu.edu.cn (Y.W.)
\* Correspondence: fenglei_han@hrbeu.edu.cn; Tel.: +86-18345152097

**Abstract:** Diver propulsion vehicles (hereinafter referred to as DPV) are a kind of small vehicle with underwater high-speed used by divers, who are able to grasp or ride on, and operate the volume switch to change the speed. Different from unmanned underwater vehicles (UUVs), the interference caused by diver's posture changing is a unique problem. In this paper, a Diver–DPV multi-body coupling hydrodynamic model considering rigid body dynamics and fluid disturbance is established by analyzing the existing DPV related equipment. The numerical simulation of multi-body articulated motion is realized by using Star-CCM+ overlapping grid and DFBI 6-DOF body motion method. Five cases of DPVs underwater cruising in a straight-line when restraining diver movement is simulated, and five cases with free diver movement are simulated too. Finally, the influence of the diver's posture changing on the cruising speed resistance is analyzed, and the motion equation including the disturbance is solved. The final conclusion is that, the disturbance is favorable at high speed, which can reduce the cruising resistance, and unfavorable at low speed, which increases the cruising resistance. The friction resistance $F_f$ always accounts for the main part in all speed cases.

**Keywords:** diver propulsion vehicle; numerical simulation; multi-body coupling; hydrodynamic interference; overlapping grid; CFD

## 1. Introduction

Diver propulsion vehicle (hereinafter referred to as DPV) is a kind of small vehicle with underwater high-speed used for divers [1], who are able to grasp or ride on, and operate the volume switch to change the speed. The numerical simulation of DPV cruising in water is different from that of HOVs (human-occupied vehicles) and UUVs (unmanned underwater vehicles). That is, when DPV cruising, the diver's body will swing with the flow field, and its posture change will have a nonlinear disturbance to the surrounding flow field, which has a huge impact on the speed and stability of the DPV. Meanwhile, the DPV's hydrodynamic shape and propeller arrangement will also have an impact on the diver body's posture. There is also the problem of the propeller wake impacting the diver's body. In the traditional numerical simulation method, diver and DPV are regarded as a relatively fixed rigid body. While in the multi-body coupling model, the joints of diver body and DPV are regarded as a complex hinge connecting rod structure, which has active and driven motion, and the two have the influence of fluid disturbance and rigid hinge force. Using CFD (compulation fluid dynamics) method to simulate the underwater cruising of a multi-body coupling model, and finally predicting the performance and analyzing the posture of the diver are the keys to solving the design optimization problem of high-performance DPVs.

Since the last century, scholars from all over the world have begun to study the design theory of DPVs and other diving equipment. In 1987, Presswood, Clark G, Godshall, David and others [2] first proposed a set of propeller matching theory applicable to DPV, and applied to the optimization of Tekna DV-3X and MIL-UNIT MOD S-5100 DPV.

CFD method is widely used in the numerical simulation of DPV linear cruising. In 2017, M.R. Sadeghizadeh and B.Daranjam [3] studied the numerical simulation of the diver–DPV coupling model by CFD in detail, and put forward the DPV optimization theory of improving the rapidity and reducing the impact on the human face is proposed. The problem that part of diver body is exposed to the external flow field is considered in the modeling. Finally, the model experiment of towing tank is carried out and the accuracy of the numerical simulation is verified, and its CFD velocity cloud chart is presented in Figure 1. In particular, the *k-ω* SST Turbulence model is used to simulate the complex vortex region at the connection between the diver's body and DPV, and the mesh of the local prism layer is encrypted on the human face.

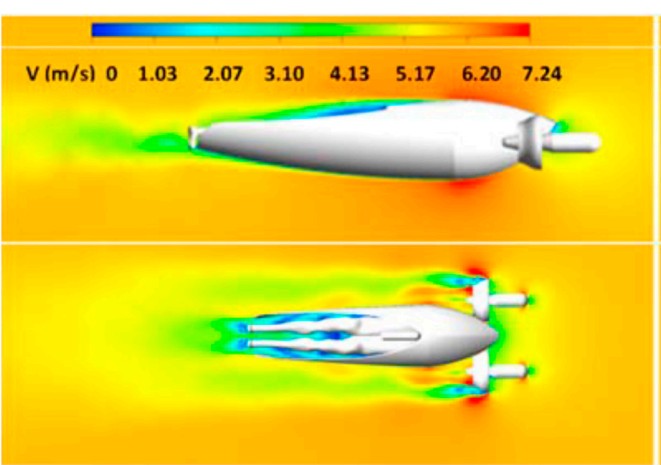

**Figure 1.** CFD velocity cloud of DPV–diver coupled model in M.R.Dadeghizadeh's research [3].

In the CFD method of DPV, the movable human body modeling is also the research focus. At the end of the 20th century, the body hydrodynamic modeling and CFD numerical simulation of swimmers and divers were began to take. In 1996, Bixler, B. S. and M. Schloder et al. [4] from University of Calgary firstly put forward CFD method to simulate the dynamic and posture model of swimmers, so as to predict, analyze, and improve the performance of swimmers.

In terms of 3D action model of swimmers, Motomu Nakashima, Ken Datou, and Yasufumi Miura [5] proposed a method for the establishment of a numerical simulation model of swimming human body considering rigid body dynamics and nonlinear hydrodynamic interference in 2007. In this model, the human limb model rotates in the water, and its rotation angle and root bending moment are measured, and the splitting steps are shown in Figure 2. Although the error of the model is 10%, it can better reflect the unsteady fluctuation of experimental data. In addition, it also simulates the position of the human body that can generate the gliding lift to determine the tangent resistance coefficient. Finally, a complete six-step freestyle simulation example is given. The simulation speed is 7.5% lower than the actual swimming speed.

CFD method of swimming human is divided into grid method and grid free method. COHEN, R.C.Z. et al. [6] proposed a computational method for human swimming using smoothed particle hydrodynamics (SPH method) in the year of 2009. It points out that the SPH method can better solve the influence of turbulence, complex free surface motion (including gas splash and entrainment) and the rapid deformation of swimmers' geometric structure. The preliminary simulation of SPH traction for male and female swimmers at different speeds with fixed gliding posture is carried out. In the experiment, laser scanning is used to establish a more accurate three-dimensional model of human body, and the dynamic model of articulation is generated based on the surface mesh deformation of human bones. Finally, the underwater video shot of swimmers is captured to verify the model.

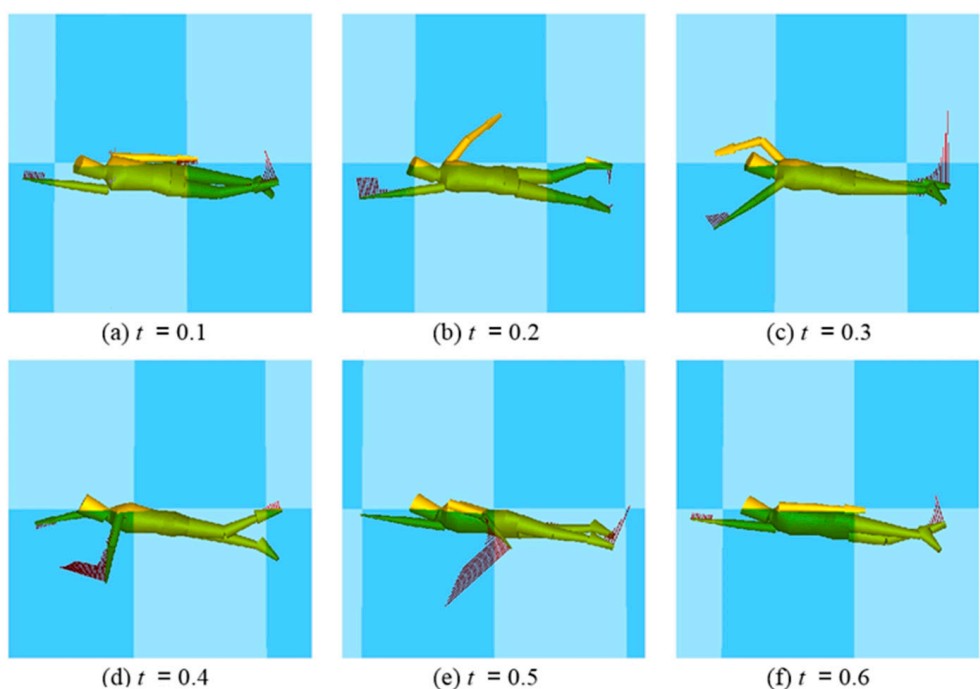

**Figure 2.** Swimming dynamics modeling and dynamic process simulation proposed by Nakashima et al. [5].

In the aspect of grid method, M.R. Sadeghizadeh and B. Daranjam et al. [7] conducted numerical simulation of three-dimensional two-phase turbulent flow on the basis of fluid volume method (VOF) of FLUENT software and tetrahedral mesh in 2012, so as to evaluate the effective power required by high-speed underwater swimmers. The speed of the swimmer model is increased to 8 m/s by using the towing tank experiment, and the results are consistent with CFD.

In 2010, Zaidi H. et al. [8] analyzed different types of turbulence models in CFD of swimmers in order to better predict resistance in fluid environment. The test of two turbulence models shows that the standard k-ω model can accurately predict the resistance and capture the vortex structure formed by the back and buttocks of underwater swimmers, while the standard $K$-$\omega$ model underestimates the value of resistance. Finally, the flow visualization experiment in insep (National Institute of Sports) verifies the rationality of vortex structure under $K$–$\omega$ model.

In 2012, Ramos et al. [9] used CFD method to analyze the influence of water depth on the resistance of swimmers during underwater gliding, and calculated the resistance coefficient (increment is 0.5 m/s) in the speed range of 1.5–2.5 m/s. It is found that the resistance decreases with the increase of depth when the center line of swimmer's modeling is in different depths of 0–0.75 m (increment is 0.25 m), while when the depth is greater than 0.75 m, the resistance value is almost the same stay the same.

Compared with swimmers, the underwater motion of diver wearing scuba diving equipment is more complex. In 2016, Darko Valenko et al. [10] firstly established the dynamic model and simulation method of scuba diver and buoyancy control device (BCD), simulated the complete heave motion process, and compared the simulation or results with experimental results.

The problem of divers driving DPV in the water is essentially the mutual disturbance of multiple underwater objects, such as the problem of the submarine running near the wall, the problem of two vehicles meeting or exceeding in narrow waters, the mutual interference between the legs of the jacket platform, and so on. Previous studies on the two-dimensional analytical solutions of multi-body perturbation problems under the condition of potential flow have been carried out in detail.

In 2010, A.A. Tchieu, D. Crowdy, et al. [11] studied the interaction between two arbitrary bodies in two-dimensional non viscous fluid, and gave the linear velocity and angular velocity of the object. For the more complex problem with divers, it can be regarded as the problem of underwater multi-body articulated motion, which has a wider application in the field of numerical simulation and experimental research of bionic fish robots. In 2018, Lijun Li, Gen Li, et al. [12] carried out numerical simulation and motion control simulation on the multi fin propulsion of puffer fish. Ida Louise G. Borlaug and Kristin Y. Pettersen proposed a generalized over twist algorithm in 2019 [13,14], which is used to track the angle, position, and direction of the base joint of the 6-DOF of AIAUV (a biotic snake AUV, shown in Figure 3).

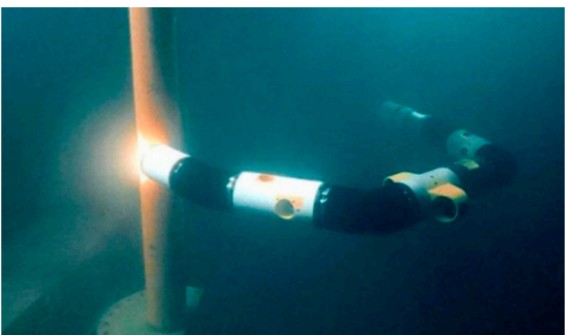

**Figure 3.** The Biotic AIAUV Eelume™ is repairing the underwater pipeline [13].

In the CFD method of multi joint biotic fish robots, the main difficulty lies in using more accurate moving grid technology and grid deformation technology to discretize the region, and using efficient solver to make the convergence speed catch up with the movement and deformation speed. Qing Xiao, Ke sun, Hao Liu, et al. [15] have studied the hydrodynamic performance of the wave NACA0012 plate near wake cylinder, which can simulate the multi-body disturbance of the rigid AUV shell to the BCF tail fin. The nephogram of vorticity is shown in Figure 4.

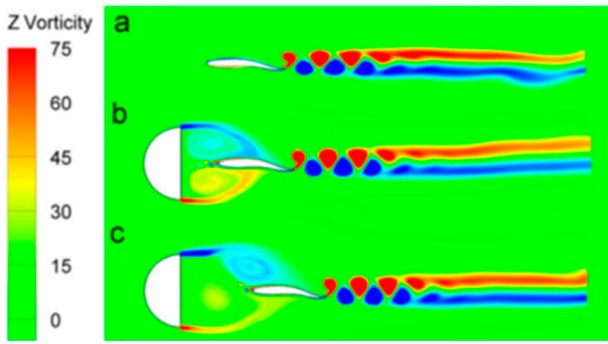

**Figure 4.** Multibody disturbance of cylinder and wing [16].

Ruoxin Li, Qing Xiao, Yuan Liu, Jianxin, et al. [16] proposed a multi-body dynamics algorithm in 2016, which combines the incompressible fluid flow with the bionic multi-body dynamics system. The tool has shown its powerful ability to solve a variety of biomechanical fish swimming problems, including self-propelled multi DOF with rigid undulator.

Yang Luo, Qing Xiao, Guangyu Shi, et al. proposed the multi body dynamic algorithm advanced code in 2019 [17] to carry out numerical simulation on the robot fish equipped with soft bionic fins. It is a fluid structure interaction (FSI) solver, which carries out finite element calculation based on CalculiX software.

In this paper, the flexible deformation of divers' fins is ignored in numerical simulation, so the function of Star-CCM+ software can be used to realize the numerical simulation of underwater multi-body articulated motion. There has been accurate CFD methods for the underwater motion of human body, which splits the swimming posture, simulates and integrates each step separately, and proposes the grid model and turbulence model with the most accurate calculation results. However, for the DPV underwater cruising, predecessors have fixed the diver body with DPV rigidly in the model, without active and passive motion of human joints. That is to say, the influence of nonlinear disturbance of multi rigid body is ignored. On this basis, the main research contents are as follows:

(1) A diver–DPV coupled model with changeable posture is established.
(2) The cruising resistance of rigid connection immobility model is solved by two CFD methods of overlapping grids and DFBI multibody motion.
(3) The influence of DPV resistance and movable human driving posture disturbance on the underwater cruising is studied.

## 2. Materials and Methods

### 2.1. Numerical Methodology

In this paper, Star-CCM+ software is used to simulate the nonlinear motion of underwater multibody. The following laws are satisfied for the Solver: conservation of mass, conservation of momentum, and conservation of energy. If the flow state is turbulent, the system also needs to follow the turbulent transport equation. It is defined as: $Re = \rho UL/\mu = UL/\gamma$, where $U$ is the velocity of simulated free flow, and $\gamma$ is the kinematic viscosity of fluid. According to the minimum length of the smallest part in this study, the minimum Reynolds number $Re_{min}$ occurs on the smallest appendage, and $Re_{min}$ is about 44005, which is shows that the flow field is turbulent, and the Reynolds number of the whole model's cruising speed $Re$ is 6816272, which belongs to high Reynolds number.

In this paper, $K$–$\omega$ SST turbulence model is adopted, which is a two-way eddy viscosity model. The formula used in the interior of the boundary layer makes the model can be directly used to the wall through the viscous sublayer. Therefore, $K$–$\omega$ SST model can be used as turbulence model with low $Re$ number and without any additional damping function. The SST formula is also converted to $K$–$\omega$ method in free flow, thus avoiding the common problem that the model is too sensitive to the turbulent characteristics of the inlet free flow. The $K$–$\omega$ SST model has good performance in adverse pressure gradient and separation flow [18].

In this paper, $K$–$\omega$ SST turbulence model is used. It meets the following requirements:
Continuity equation:

$$\vec{\nabla} \cdot \vec{u} = 0 \tag{1}$$

where $\vec{\nabla}$ is Hamilton operator, $\vec{u}$ is velocity vector.
Momentum equation:

$$\begin{aligned}
\frac{\partial}{\partial x_j}(\rho u_i u_j) &= -\frac{\partial P}{\partial x_i} + \frac{\partial t_{ij}}{\partial x_i} + \frac{\partial \tau_{ij}^t}{\partial x_i} \\
t_{ij} &= \mu\left(\frac{\partial u_i}{\partial x_j} + \frac{\partial u_i}{\partial x_i}\right) \\
\tau_{ij}^t &= -\rho \overline{u'_i u'_j}
\end{aligned} \tag{2}$$

where $P$ is the preesure, $t_{ij}$ is the stress tensor, $\tau_{ij}^t$ is Reynolds stress term. $u_i$, $u_j$ is the fluctuating velocity component in the $i$, $j$ direction fluid density, $\mu$ is the fluid dynamic viscosity, and $\rho$ is the fluid density.

Because in turbulence, the actual velocity of each particle is the sum of instantaneous and fluctuating components. The last term $\overline{\rho u'_i u'_j}$, the Reynolds stress, should be obtained using a suitable turbulence model.

Predecessors have proved which model is more effective than other models with the same conditions. For example, Zaidi et al. (2010) proposed that $k$-$\omega$ SST model is more

accurate for predicting human underwater swimming resistance. According to their model, $k$ and $\omega$ are

$$\frac{\partial}{\partial t}(\rho k) + \frac{\partial}{\partial x_i}(\rho k u_i) = \frac{\partial}{\partial x_j}(\Gamma_k \frac{\partial w}{\partial x_j}) + G_k - Y_k + S_k$$
$$\frac{\partial}{\partial t}(\rho \omega) + \frac{\partial}{\partial x_i}(\rho \omega u_i) = \frac{\partial}{\partial x_j}(\Gamma_w \frac{\partial \omega}{\partial x_j}) + G_\omega - Y_\omega + S_\omega \tag{3}$$

where the parameter "$\omega$" is the energy dissipation rate (or specific dissipation rate) per unit volume, and "$k$" is the turbulent kinetic energy per unit mass. $\Gamma_k$ and $\Gamma_w$ are the effective diffusivities of $k$ and $\omega$. $G_k$ and $G_w$ are the turbulence results of $k$ and $\omega$. $Y_k$ and $Y_\omega$ are the turbulent dissipations of $k$ and $\omega$. $S_k$ and $S_\omega$ are source terms of $k$ and $\omega$.

Therefore, the resistance of DPV can be obtained by this method, and the propulsion power $P_p$ related to the required cruising speed $V$ can be calculated as

$$P_p = F_t \cdot V \tag{4}$$

where $F_t$ is real time resistance of hull.

On the other hand, if the propulsion power is electric, other parameters will come into question, such as propulsion efficiency $\eta_p^-$ (composed of propeller and motor efficiency) and the total weight of diver and DPV. Under this condition, the input power $P_i$ is defined as the formula

$$P_i = \frac{P_p}{\eta_p^-} \tag{5}$$

### 2.2. 6-DOF Body Splitting and Model Building

Firstly, the hydrodynamic shape of DPV is modeled. Whether DPV or AUV, its hydrodynamic shape directly affects the resistance and posture of underwater cruising. In this paper, a streamlined DPV with pump-jet propulsion is designed as the research object, which is referring to the Rotinor®Blackshadow 730 DPV [19]. Its parameters are in Table 1.

**Table 1.** Parameters of four typical DPV

| Main Indexes | Notations | Value |
|---|---|---|
| Length | $L_{DPV}$ | 950 mm |
| Width | $B_{DPV}$ | 400 mm |
| Height | $H_{DPV}$ | 300 mm |
| Displacement | $m$ | 12 kg (neutral buoyancy) |
| Maximum cruise speed | $V_5$ | 2.5 m/s |
| Maximum diving depth | $D_{max}$ | 50 m |

The model abandons the exposed propeller layout of the normal DPV and adopts pump jet propulsion [20]. Which is, the inflow enters from the bottom-front openings, accelerates through the propeller in the duct, and ejects at a high velocity. In this model, when calculating the resistance of DPV, the model of propeller only retains the hub part.

In this model, the posture of diver driving DPV is considered as drag-and-drop. Which is, the diver navigates by holding the operating lever of the DPV with both hands, and uses torque to change the head direction of the DPV to carry out oblique navigation and steering movement like typical SUEX DPV [21].

This model assumes that the X-direction motion of DPV is locked, and a fixed velocity X-direction flow is used to simulate cruising. The local coordinate system was established with the centroid point $P_0$ of the hip part of diver. At this time, the typical cruising posture and force situation of diver and DPV are shown in Figure 5. The equation of motion is

$$ma_x = F_t - F_f \tag{6}$$

$$ma_y = G - F_b - F_l \tag{7}$$

where $a_x$ and $a_y$ is the acceleration in X, Y directions.

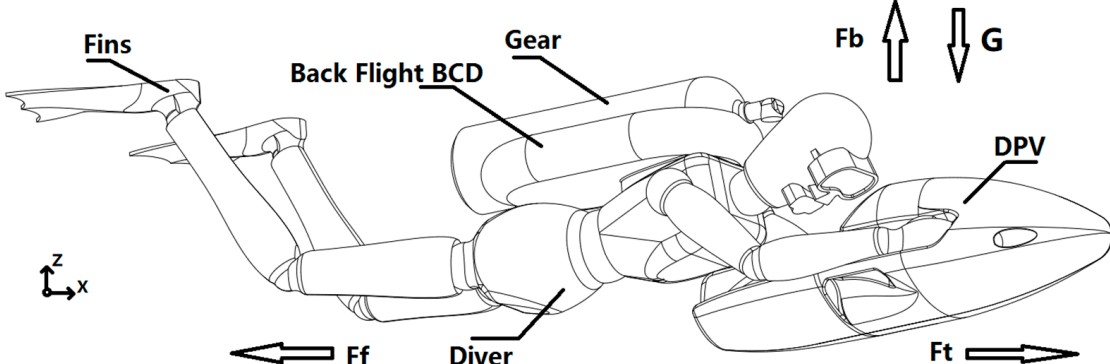

**Figure 5.** Forces on diver–DPV coupled model.

For the convenience of calculation, the diver body posture at design speed is given as $T_0$ posture. According to the Professional Association of Diving Instructors (PADI) DPV driving course, the typical driving posture should be: DPV, diver's trunk and fins are parallel to the cruising direction, DPV is parallel to *x*-axis; thighs are slightly upward bending; knees and elbows are bending at a certain angle; forearms are parallel to DPV. After quantification, the model in Figure 6 is obtained, in which the angles are assumed to be the positive direction of acute angle with *x*-axis.

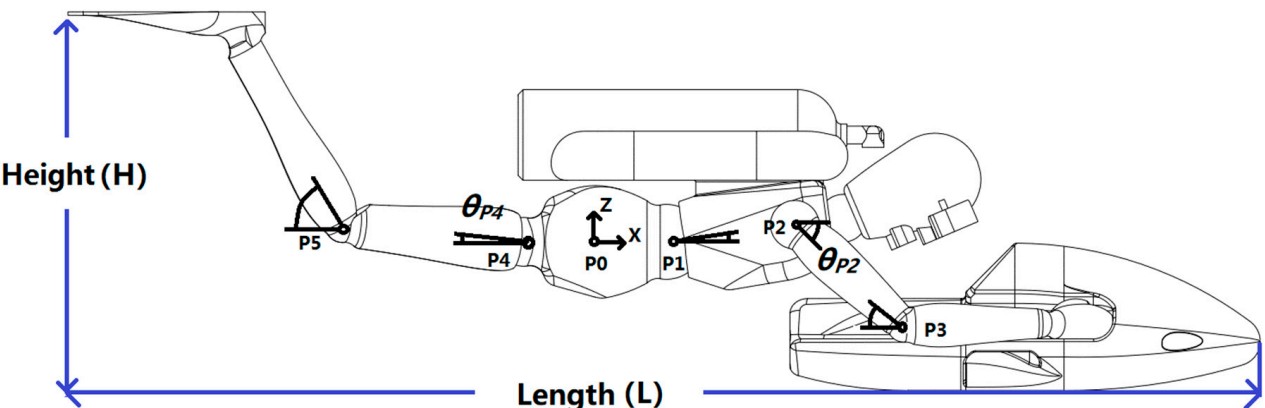

**Figure 6.** Notations of model's dimension and posture.

That is, $\theta_{p1}$ is the acute angle between the body's central axis and the *x*-axis; $\theta_{p1}$ is the acute angle between the body's central axis and the *x*-axis; $\theta_{p2}$ and $\theta_{p3}$ is the acute angle between the upper arm's central axis and the *x*-axis, and $\theta_{p3} = \theta_{p2}$; $\theta_{p4}$ is the acute angle between the thigh's central axis and the *x*-axis; $\theta_{p5}$ is the acute angle between the crus's central axis and the *x*-axis;

In order to make the hydrodynamic modeling of this paper quantitative and more universal, the dimensions and driving posture of four typical DPV as SEADOO, U.FLIGHT, SUEX, ROTINOR are analyzed through image data Figure 7, and the following Table 2 is obtained.

In this paper, a typical diver's body is selected, equipped with back-flying BCD (Buoyancy Control Device) and open breathing apparatus [23]. According to Table 1, the posture angles of DPV with different models and speeds are slightly different. Since the shape and max cruising speed of DPV in this paper are similar to ROTINOR, the $T_0$ posture angle is selected as Table 3.

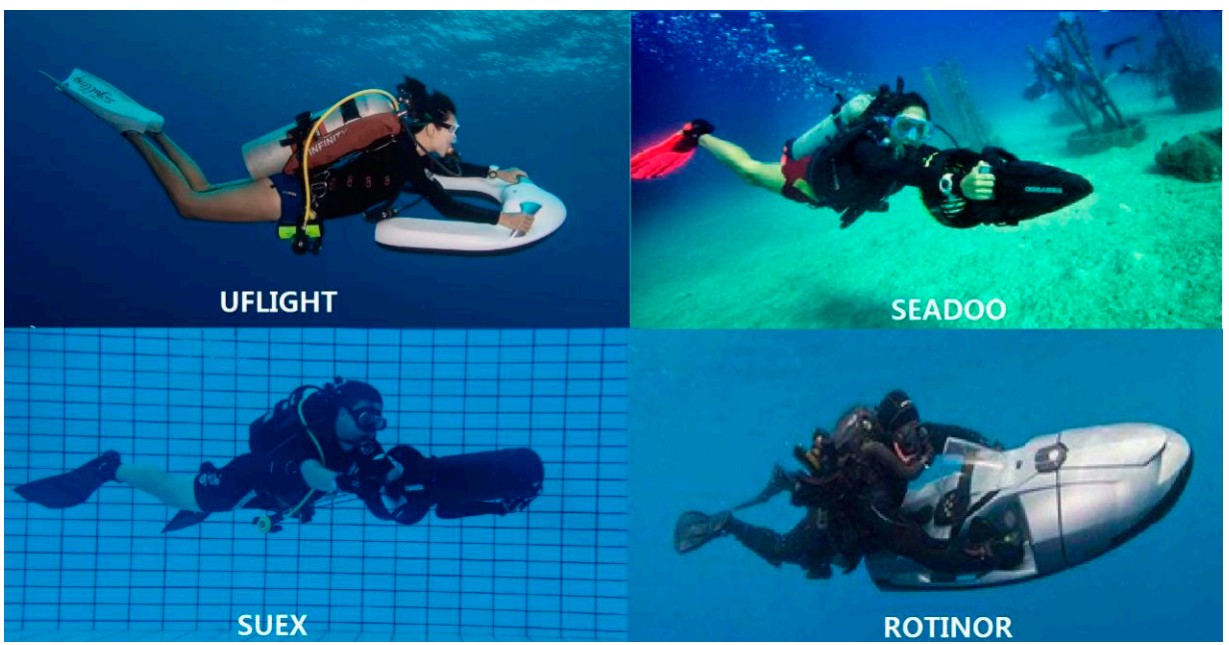

**Figure 7.** Driving posture of four typical DPV [19,21,22].

**Table 2.** Parameters of four typical DPV

| DPV Type | Length of DPV $L_{DPV}$ (m) | Cruising Speed $V$ (m/s) | Body $\theta_{P1}$ | Elbow $\theta_{P3}$ | Thigh $\theta_{P4}$ | Knee $\theta_{P5}$ |
|---|---|---|---|---|---|---|
| SEADOO [22] | 0.55m | 1 | −26.6° | 37.3° | 11° | 38.6° |
| SUEX [21] | 0.814m | 1.5 | −26° | 41° | −4.5° | 52° |
| U·FLIGHT | 0.95m | 2 | −25.9° | 43.5° | −4.5° | 47.8° |
| ROTINOR [19] | 1.766m | 2.5 | −7.3° | 42.4° | 5.4° | 60.9° |

**Table 3.** Posture angle at $T_0$ moment

| Definitions | Notations | Value |
|---|---|---|
| Length of DPV | $L_{DPV}$ | 960 mm |
| Body posture | $\theta_{p1}$ | 7.3° |
| Shoulder posture | $\theta_{p2}$ | 42.4° |
| Elbow posture | $\theta_{p3}$ | 42.4° |
| Thigh posture | $\theta_{p4}$ | 5.4° |
| Knee posture | $\theta_{p5}$ | 60.9° |

In addition, the elbow joint angle $\theta_{p3}$ of all the four DPV is within 12%, and the knee joint angle $\theta_{p5}$–$\theta_{p4}$ of the other three DPV is within 7.4% except SEADOO. Therefore, in order to further facilitate the calculation, the model is simplified as follows:

(1)　The angle of knee joint and elbow joint remained unchanged at $T_0$;
(2)　All joints only rotate around *y*-axis;
(3)　The DPV is rigidly fixed to the forearm and parallel to the *x*-axis when cruising;
(4)　All solids have the same density as water

According to the above conditions, the diver–DPV coupling model are divided into three articulated 6-DOF rigid bodies. As shown in Figure 8, they are named DPV and arms, body and hip, and legs respectively. The hinge points of them are P2 and P4, and the quality attributes are shown in Table 4. Three reference coordinate systems of them named $O_1$-$X_1Y_1Z_1$, $O_2$-$X_2Y_2Z_2$, $O_3$-$X_3Y_3Z_3$ are established. The symbols for the force components are defined in Figure 8.

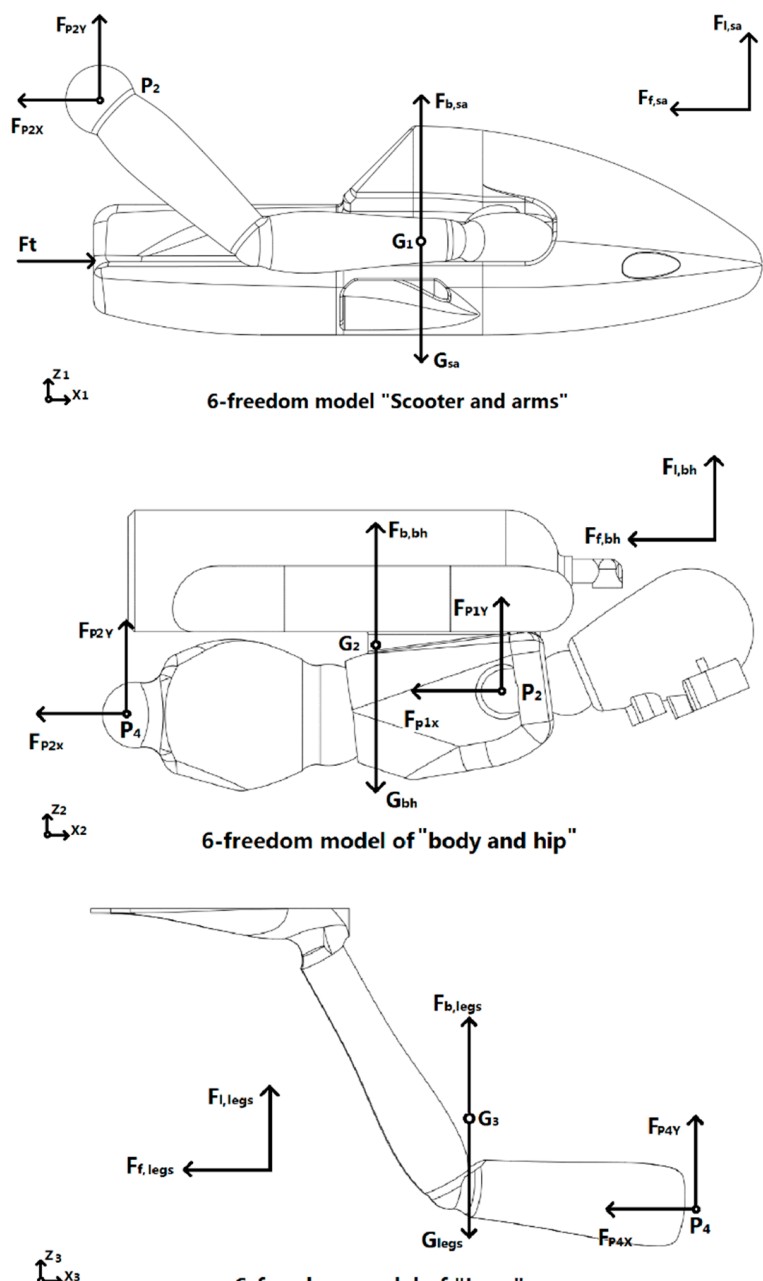

**Figure 8.** Simplified three 6-DOF rigid bodies.

**Table 4.** Mass properties of three 6-DOF bodies.

| Model | G (kg) | G (x,z) | $I_x$ (kg·m$^2$) | $I_y$ (kg·m$^2$) | $I_z$ (kg·m$^2$) |
|---|---|---|---|---|---|
| DPV and arms | 42.13 | 840.232 −171.604 | 0.643297 | 2.539773 | 2.733827 |
| Body and hip | 53.48 | 227.264 114.077 | 0.822940 | 3.110566 | 3.404638 |
| Legs | 13.8 | −0.515 0.155 | 0.471254 | 1.033288 | 1.409276 |

### 2.3. Force Components and Disturbance Analysis

The causes of the external force components are explained as follows: According to the driving principle of DPV, when the propeller is started, the DPV generates axial thrust $F_t$, which is transmitted to the diver's arms. At the shoulder hinge point P$_2$, $F_{P2Y}$, and

$F_{P2X}$ are applied to the body and hips. At the same time, the DPV and arms are subjected to hydrodynamic forces, which are decomposed into $F_{l,da}$ and $F_{f,da}$. Body and hips, and legs are also subject to the tension at the hinge point and hydrodynamic forces, and the fluid-solid coupling and solid–solid coupling between multi rigid bodies exist in it [24].

Next, the coupling disturbance components are analyzed: for DPV and arms, in addition to the theoretical value calculated from hydrodynamic coefficient, the disturbance generated in the flow field due to the motion and posture of body and hip, and legs need to add a disturbance term $F_{dis1}$. According to the external force analysis of the 6-DOF body in the Figure, the disturbance terms of DPV and arms in the $x$-axis and $z$-axis directions of geodetic coordinate system 0-XYZ at the initial T0 time are obtained as

$$m_{da}a_{Z,da}(t) = G_{da} - F_{b,da} - F_{P2y} - F_{dis1,y} \tag{8}$$

where $G_{da} = m_{da}g$, $F_{b,da} = \rho_w g V_{da}$, $F_{f,da} = 1/2 C_{L,da} \rho_w v(t)^2 S_{da}$

$$m_{da}a_{X,da}(t) = F_t - F_{P2x} - F_{f,da} - F_{dis1,x} \tag{9}$$

where $F_{f,da} = 1/2 C_{t,da} \rho_w v(t)^2 S_{da}$

$C_{L,da}$ is the lift coefficient, $\rho_w$ is the density of water, $F_P$ is the hinge point force between the 6-DOF bodies, $F_b$ is the buoyancy of the 6-DOF bodies, $G$ is the gravity of the 6-DOF bodies, $F_t$ is the propeller trust, $F_{dis}$ is the disturbing force of the other two 6-DOF bodies. Similarly, for body and hip

$$m_{bh}a_{Z,bh}(t) = G_{bh} + F_{P2y} - F_{b,bh} - F_{P4y} - F_{dis2,y} \tag{10}$$

$$m_{bh}a_{X,bh}(t) = F_{P2x} - F_{f,bh} - F_{dis2,x} - F_{P4x} \tag{11}$$

for legs, we get

$$m_{legda Z,legs}(t) = G_{legs} + F_{P4y} - F_{b,legs} - F_{dis3,y} \tag{12}$$

$$m_{legda X,legs}(t) = F_{P4x} - F_{f,legs} - F_{dis3,x} \tag{13}$$

To sum up, when the DPV curise at a uniform speed, the acceleration $a$ is zero. Combined with the above formula, along the $x$-axis of the geodetic coordinate system, there are

$$F_t = F_f = F_{f,da} + F_{f,bh} + F_{f,legs} + F_{dis1,x} + F_{dis2,x} + F_{dis3,x} \tag{14}$$

where $F_{f,da} + F_{f,bh} + F_{f,legs} = F_{f0}$. That is, the resistance in the direction of $x$-axis is restrained. $F_{dis1,x} + F_{dis2,x} + F_{dis3,x} = F_{dis,x}$. That is, the posture disturbance term in the $x$-axis direction. In the fourth section, by solving and discussing the size and direction of $F_{dis}$, the disturbance factors caused by diver motion are determined.

### 2.4. Computational Region and Meshing

Firstly, the overlapping mesh is generated. In the common CFD calculation of underwater vehicle, the background region is usually set as the water tank involved in. While in order to save computational resources, the small area around the model and the area of its wake flow always set finer grid, which is called the encryption region, and the other flow area uses thicker grid.

In order to fully develop the incoming flow and avoid the interference from the boundary of the external field, the size of the background region (water tank) is adjusted to be 4-times the total length of the diver–DPV coupled model, 2.5-times of the full width, and 5-times of the full height. Furthermore, the encryption grid area is set to increase the calculation accuracy. After integer, the size is as follows in Table 5.

**Table 5.** Dimensions of three regions.

| Region | Length, *L* (m) | Width, *B* (m) | Heigth, *H* (m) |
|---|---|---|---|
| Model | 2.429 | 0.843 | 0.764 |
| Encryption | 4.000 | 1.300 | 2.400 |
| Water tank | 9.000 | 2.000 | 4.000 |

The side length of the grid in the water tank region is 10cm, and the side length of the grid in the encryption area is 1cm (0.4% of the model length). The grid number of water tank is 6704321, and that of the encryption is 1052876. Finally, the calculated areas are shown in Figure 9.

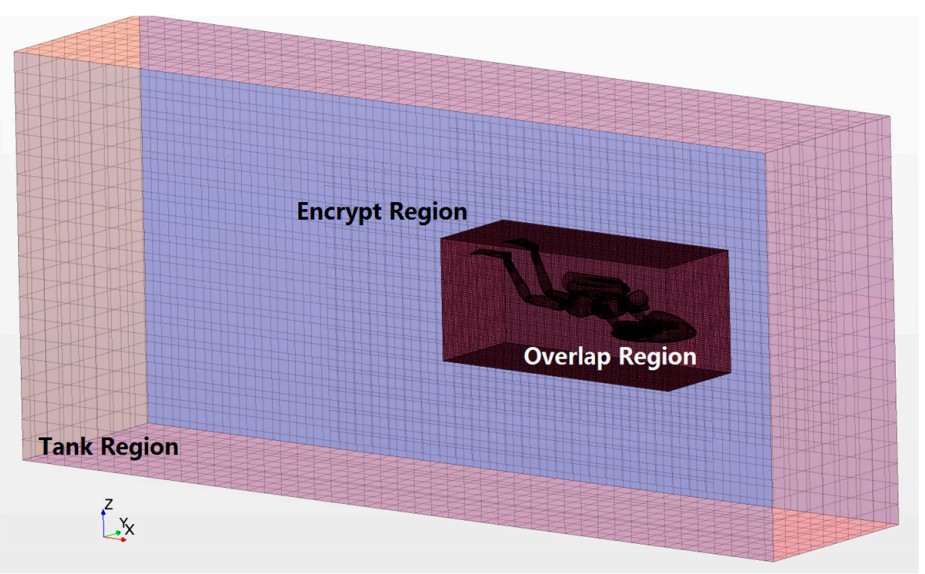

**Figure 9.** Calculation regions of water tank.

In order to simulate the disturbance of diver joint motion on the hydrodynamic performance simulation calculation, the following provisions are made on the basis of region division:

(1)  The rigid body of DPV and arms is bound on the water tank region, that is, the region moves together with the motion of DPV, which can ensure that the DPV will not run out of the water tank during cruising;

(2)  The dynamic region is set based on Body and Hip and Legs. In order to improve the calculation accuracy, its contour envelops the whole diver–DPV model, which is shown in Figure 10.

(3)  In particular, the overlapping grid relationship is established between the surfaces of the three overlapping mesh continuum to ensure that the fluid converges when passing through the interface.

(4)  The encrypted region is set large enough to ensure that the overlapped grid is always included in when in motion, so as not to diverge due to different scales after contacting with the external flow field.

Finally, the side length of the overlapped grid region is also 1cm (the same as the encryption region), and the overlapped grid number of body and hip, and legs are 492833 and 462466. The number of grids in all regions is with a total of 7659620.

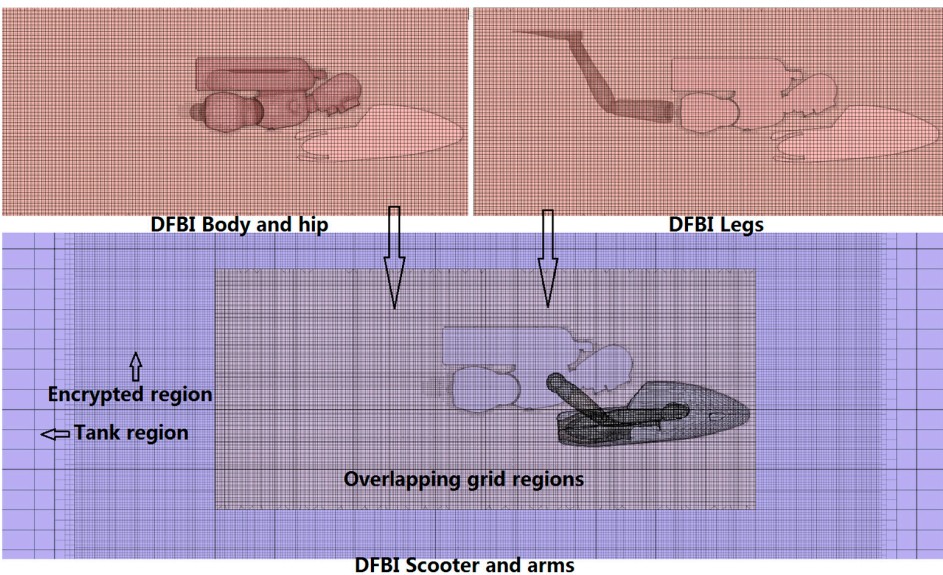

**Figure 10.** Overlapping grid and encryption region.

After all the regions are established, the DFBI (Dynamic Fluid Body Interaction) motion function of Star-CCM+ software is used to make the 6-DOF body produce constrained motion. The DFBI body model can simulate the passive motion of rigid body under the action of fluid. That is, the DOF of the rigid body can be determined according to the needs, and the specific DOF can be constrained/released. The mass, moment of inertia, and centroid coordinates of the three rigid bodies of diver–DPV model are described in Table 2, while the specific degrees of freedom constraints of each DFBI body are described in Section 3.2. The joint and shoulder coordinate are needed to be supplement:

$$P_2 \ (-0.455196 \text{ m}, 0.0 \text{ m}, 0.234889 \text{ m}) \ P_4 \ (-0.3711 \text{ m}, 0.0 \text{ m}, -0.12 \text{ m})$$

The boundary conditions of regions are set as follows: The upper, lower and the inflow surface of the water tank region are set as the velocity-inlet, the back surface is the pressure-outlet, and the left- and right-side surface are set as symmetrical-planes. An overlapping grid boundary is established between the outer surface of the leg region, the body region, and the water tank region, while the model surfaces in the three overlapping regions are set as the wall-surface. The $k$–$\omega$ SST turbulence model is used to close the N-S equation. The full $y+$ wall function method suitable for high Reynolds number is used for wall treatment. First boundary layer mesh is obtained by $y+=30$, and the case $y+$ distribution is shown in Figure 11. Finally, 10 layers of prismatic mesh are adopted, and the thickness of the first prismatic mesh is 0.0073 mm according to $y+$.

In the aspect of solver, the time step is selected first. When the time step is too small, the calculation time is too long, and when the time step is too large, the calculation accuracy is not enough. The resistance time step is generally 0.005 L/V~0.01 L/V, where $V$ is the velocity, and the corresponding Courant number of this model is 0.2–0.4. Therefore, 0.005 s is taken as the time step, which satisfies the CFL condition. Set the maximum number of internal iterations to 5. The SIMPLE (Semi Implicit Method for Pressure Linked Equation) algorithm is used to solve the Navier–Stokes equations. The viscous term in the discrete equation adopts the second-order central difference scheme, and the convective term adopts the second-order upwind scheme.

In terms of initial conditions, assuming that the experimental condition is infinite water depth, the fresh water temperature is 24 °C, $g = 9.81$ m/s$^2$, and the kinematic viscosity coefficient $v = 0.9167 \times 10^6$ m$^2$/s. The necessary DOF of the rigid bodies are constrained, and the $x$-axis cruising is simulated by uniform inflow, that is, the absolute motion is replaced by relative motion to save calculation time.

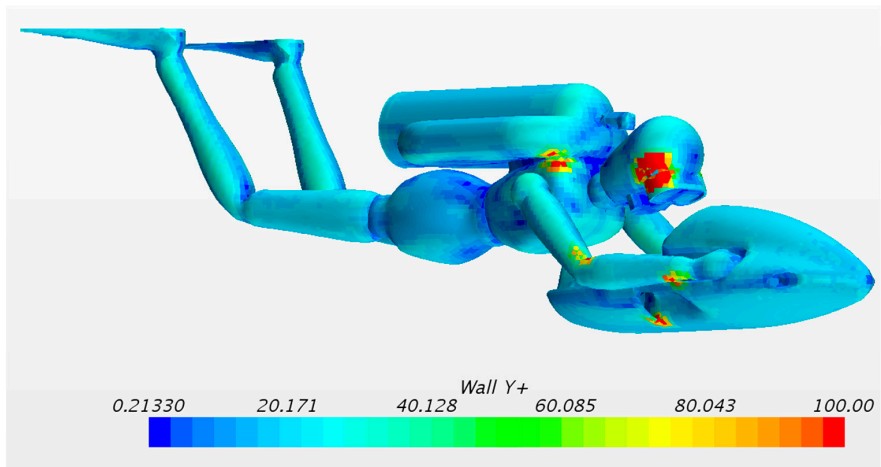

**Figure 11.** The case *y*+ distribution nephogram.

## 3. Results

This section shows the CFD results the straight-line cruising motion of the diver–DPV coupling model in the water. That is, the motion in the *x*-axis direction is simulated through the uniform incoming flow, and the translation movement in the *z*-axis direction is constrained. Firstly, the diver and DPV are regarded as a rigid body by restraining the human motion; then the human motion is released as a multi-rigid-articulated model; finally, the diver–DPV disturbance components are analyzed by comparison.

### 3.1. Case of Restricting Human Motion

This section only simulates constrained straight-line cruising. All DOF of the three 6-DOF models are fixed and restrained.

Firstly, mesh independence and convergence are analyzed by taking the incoming velocity with the DPV's maximum speed $V_5$ = 2.5 m/s as an example. The sub relaxation factor of velocity and pressure is 0.5. After 10000 time steps of simulation, the convergence result of resistance $F_f$ is shown in Figure 12. The velocity nephogram is shown in Figure 13.

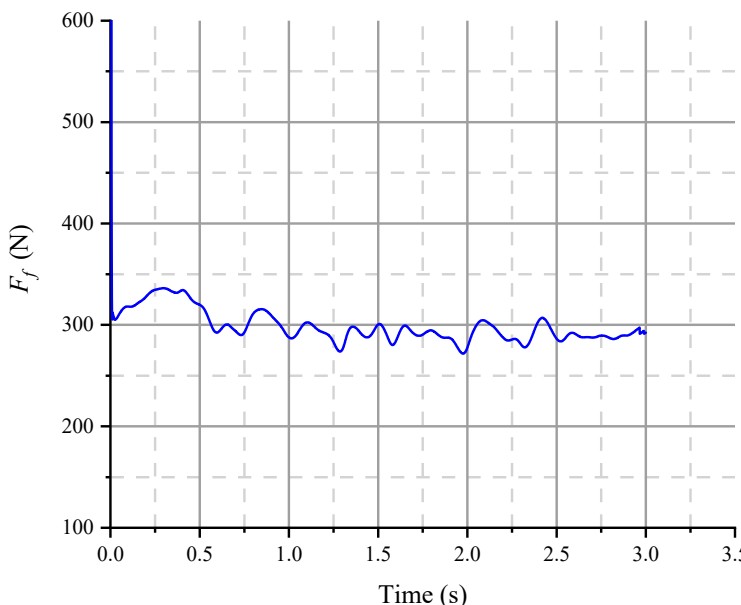

**Figure 12.** Resistance convergence results of constrained straight-line cruising simulation.

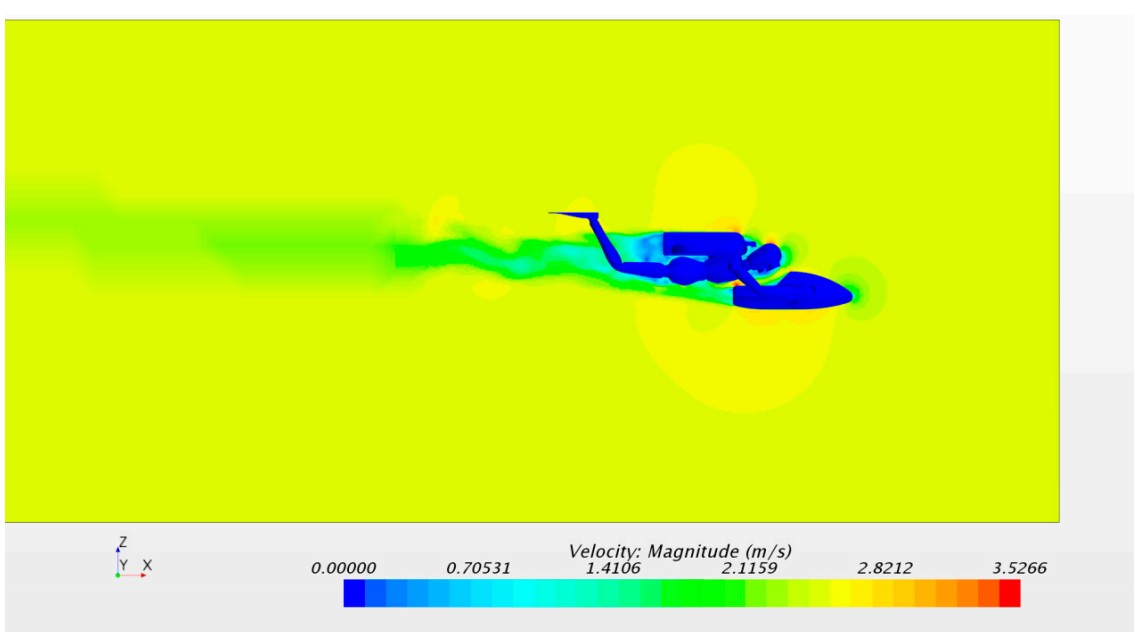

**Figure 13.** Velocity nephogram at 3s moment of constrained straight-line cruising simulation.

Then the straight-line thrust $F_t$ required by $V_5$ = 2.5 m/s is obtained. Because of the resistance tends to converge after 1s, the resistance value in the period of 1–3 s is taken as the arithmetic average. That is, $F_f = F_t$ = 300.4 N, $z$-axis lift $F_l$ = −14.2 N.

Next, the resistance prediction of constrained straight-line cruising under $V_5$ = 2.5 m/s is selected for grid independent. According to ITTC (International Towing Tank Conference) [25], the grid side length is set as 1/2000–5/2000 of the model length $L$, which is 1 cm in this paper. There set three different length as 1 cm, $\sqrt{2}$ cm, and 2 cm to generate coarse mesh. The total number of meshes is 7659620, 2708494, and 957453. When the other parameters and settings are the same, the change trend of resistance value with the foundation size is observed as Figure 14.

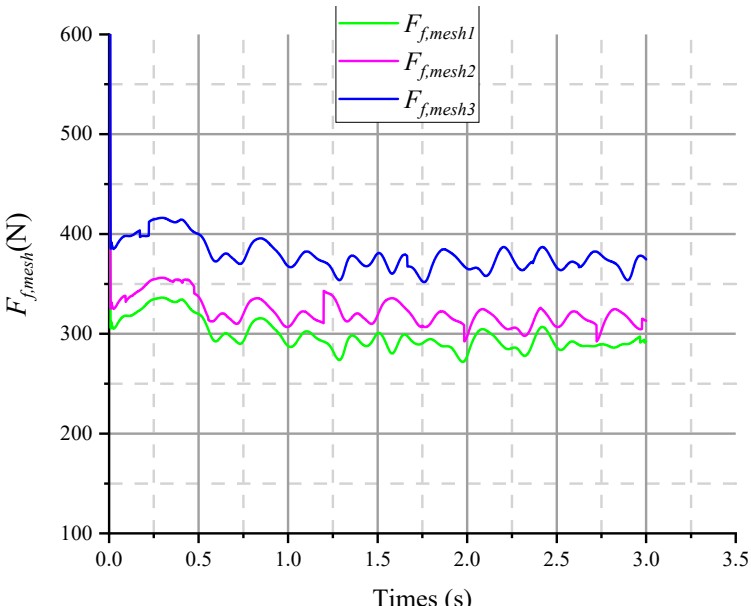

**Figure 14.** Convergence values of resistance under three kinds of grids.

Finally, the mean resistance $F_{f,mesh1}$ = 300.4 N, $F_{f,mesh2}$ = 321.2 N, $F_{f,mesh3}$ = 378.1 N. So there are:

$$\frac{F_{f,mesh3}}{F_{f,mesh2}} = 1.18 > 1.07 = \frac{F_{f,mesh2}}{F_{f,mesh1}} \tag{15}$$

Therefore, with the gradual refinement of the grid, the numerical simulation value of resistance gradually decreases, and the calculation accuracy gradually converges as 6.9%, and the number of grids increases sharply as 180%. Considering the calculation accuracy and cost, it is reasonable to take the grid side length of 1 cm.

Then, the numerical simulation of constrained straight-line cruising under the other four speed conditions is carried out. The four inflow velocities are set as $V_1$= 0.5 m/s, $V_2$ = 1.0 m/s, $V_3$ = 1.5 m/s, and $V_4$ = 2.0 m/s. So the linear relationship between $V_n^2$ and $F_f$ is obtained. The resistance coefficient $C_t$ is introduced here. Because the rigid body under water straight-line cruising has $F_f = 1/2C_t\rho_w v^2 S$, the ideal value $C_{t0}$ can be obtained from the slope of the fitting curve. After numerical simulation, the relationship between resistance $F_f$ and $V^2$ is shown in Figure 15.

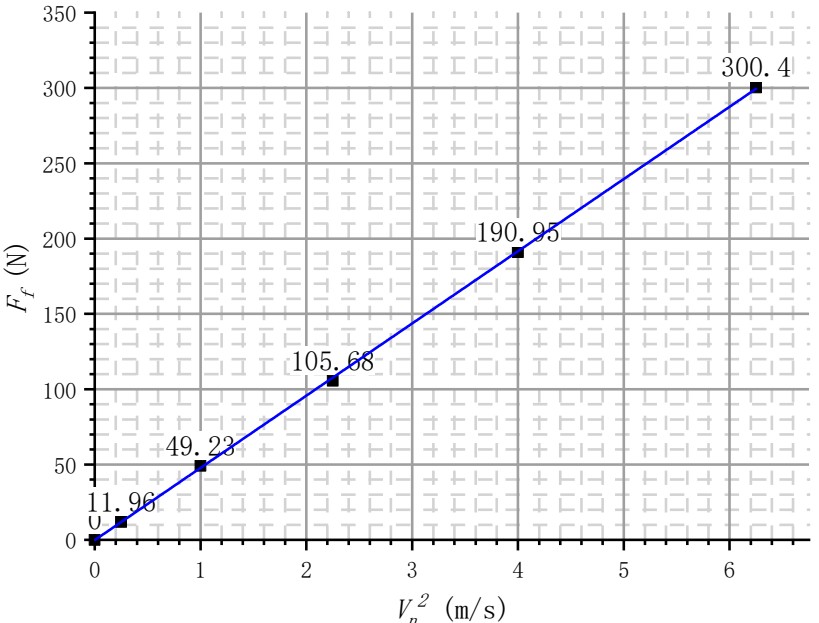

**Figure 15.** Relation curve between $V_n^2$ and $F_f$.

Finally, the resistance coefficient of diver–DPV model is obtained as $C_{t0}$ = 27.2984/10$^{-3}$.

### 3.2. Case of Releasing Human Motion

This section only carries out the simulation of straight-line cruising with released human motion. That is, releasing the *x*- and *z*-axis translation DOF and the rotation DOF around the *y*-axis of two 6-DOF bodies of body and hips, and legs to simulate the articulated motion of diver body in water.

### 3.2.1. Maximum Cruising Speed

Firstly, the numerical simulation of the maximum cruising speed $V_5$ = 2.5 m/s is carried out. The velocity clouds at the moment of 2 s is shown in Figure 16. And the variation curve of resistance $F_f$ and lift force $F_l$ are shown in Figures 17 and 18. The posture angle changes of 6-DOF model legs' $\theta_1$, and body and hip's $\theta_4$ are obtained as shown in Figures 19 and 20.

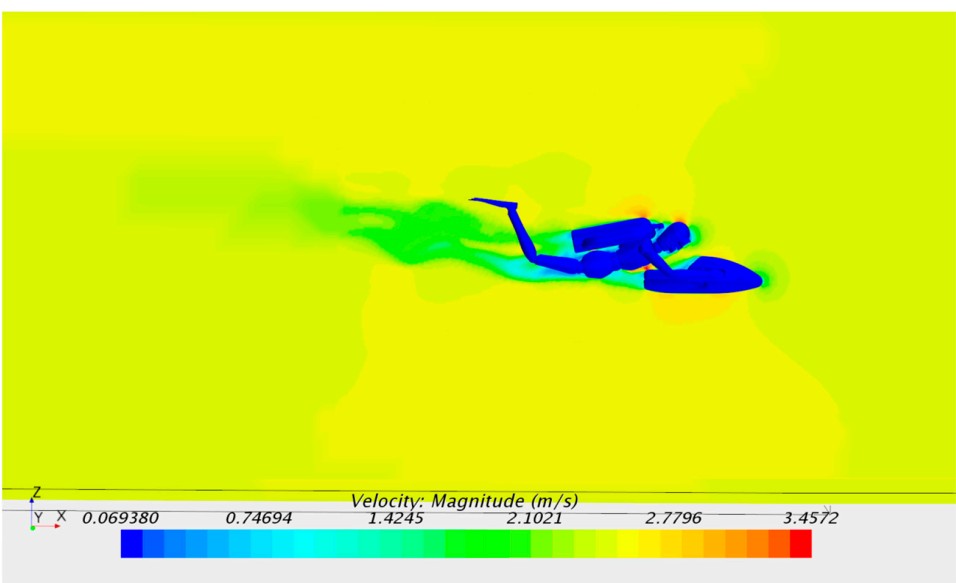

**Figure 16.** Velocity nephogram at 2 s moment of straight-line cruising with releasing motion at $V_5$ speed.

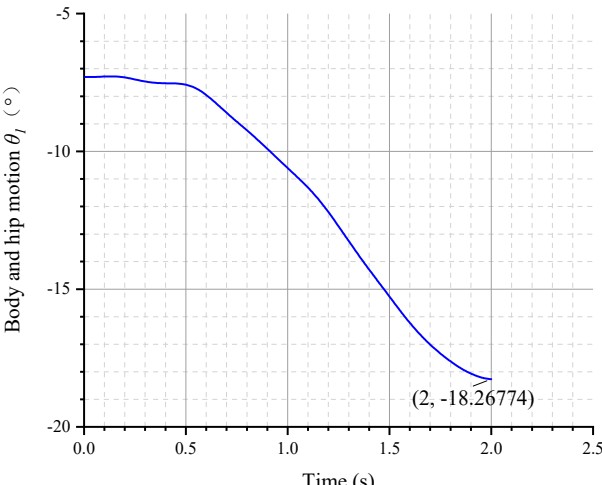

**Figure 17.** Change process of 6-DOF model body and hip posture angle.

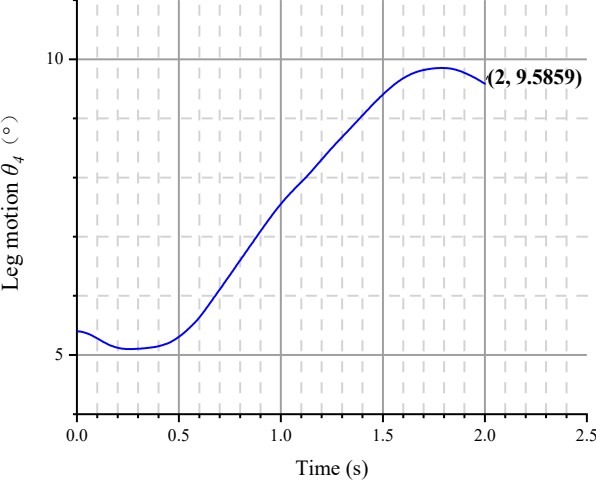

**Figure 18.** Change process of 6-DOF model legs posture angle.

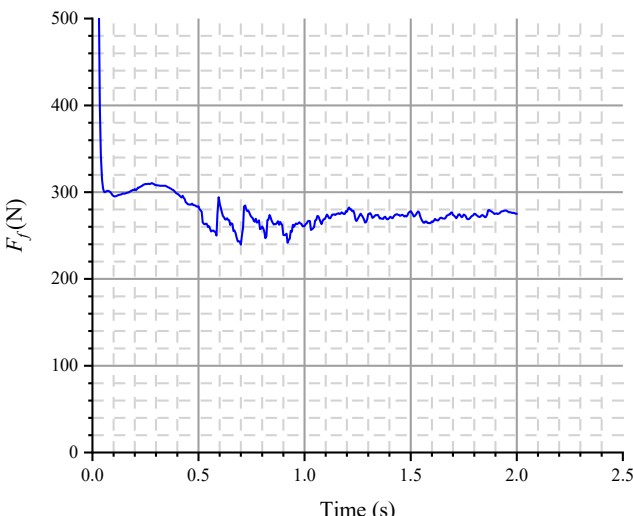

**Figure 19.** Convergence process of cruising resistance.

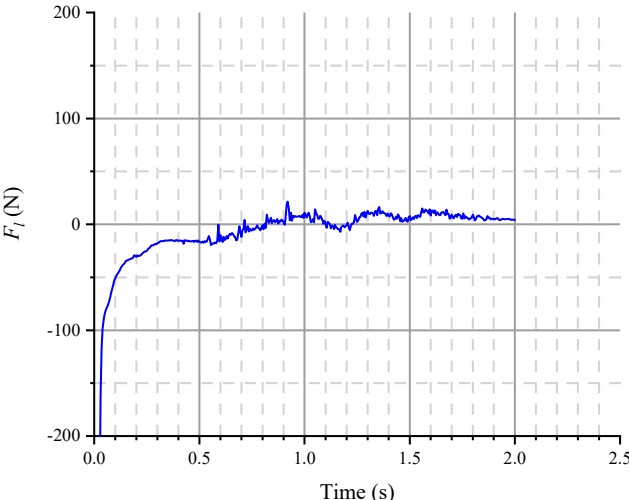

**Figure 20.** Convergence process of cruising lift.

The analysis shows that the angle $\theta_{P1}$ of Body and Hip gradually converges to the stable range, and $\theta_{P4}$ of legs first fluctuates in a certain range, and finally converges to the stable range. That is, the diver body will swing up and down during cruising. At the same time, the resistance also fluctuates in a certain range. Because the resistance converge after 1 s and fluctuates in a small range, and the human body posture reached the inflection point angle at 2 s, the resistance value in the period of 1–2 s is taken as the arithmetic average. That is, $F_{f,V5}$ = 272.72 N, and the lift $F_{l,V5}$ finally converges near 0.

### 3.2.2. Other Cruising Speeds

Next, the numerical simulation of other speeds is carried out. The *x*-axis and *z*-axis translation and rotation around *y*-axis of body and hip, and legs are released, and the *z*-axis translation of DPV and arms is restrained. Set five inflow velocities as $V_1$ = 0.5 m/s, $V_2$ = 1.0 m/s, $V_3$ = 1.5 m/s, $V_4$ = 2.0 m/s, and $V_5$ = 2.5 m/s, then the posture angle of diver body and its influence on the speed of straight-line cruising are obtained.

In fluid mechanics, a dimensionless variable characterizing the relative magnitude of inertial force and gravity of fluid is called Froude number *Fr*, which represents the ratio of the inertial force to the magnitude of gravity. That is, $Fr = v^2/gL$, where *v* is the straight-line cruising speed of rigid body, *g* is the acceleration of gravity, *L* is the characteristic length of the object.

In this paper, the resistance of the other four cases of different speeds are calculated according to *Fr* number from low to high. After CFD convergence, the mean resistance, mean lift and posture are shown in Table 6, and the velocity scalar nephogram is shown in Figure 21. The instantaneous value of posture angle is obtained at 2 s moment, and the resistance is the average value of 1~2 s.

**Table 6.** DPV resistance and posture angle at different speeds.

| *Fr* | $V_n$ (m/s) ($n = 1$~5) | Body and Hip $\theta_{P1}$ (°) | Legs $\theta_{P4}$ (°) | $F_f$ (N) | $F_l$ (N) |
|---|---|---|---|---|---|
| 0.00867 | 0.5 | −30.24 | 13.80 | 18.68 | −0.54 |
| 0.03469 | 1.0 | −28.42 | 13.30 | 52.64 | −0.66 |
| 0.07944 | 1.5 | −27.07 | 13.35 | 100.30 | 1.55 |
| 0.13875 | 2.0 | −26.32 | 14.00 | 185.13 | 1.03 |
| 0.21680 | 2.5 | −18.27 | 9.59 | 272.72 | 5.49 |

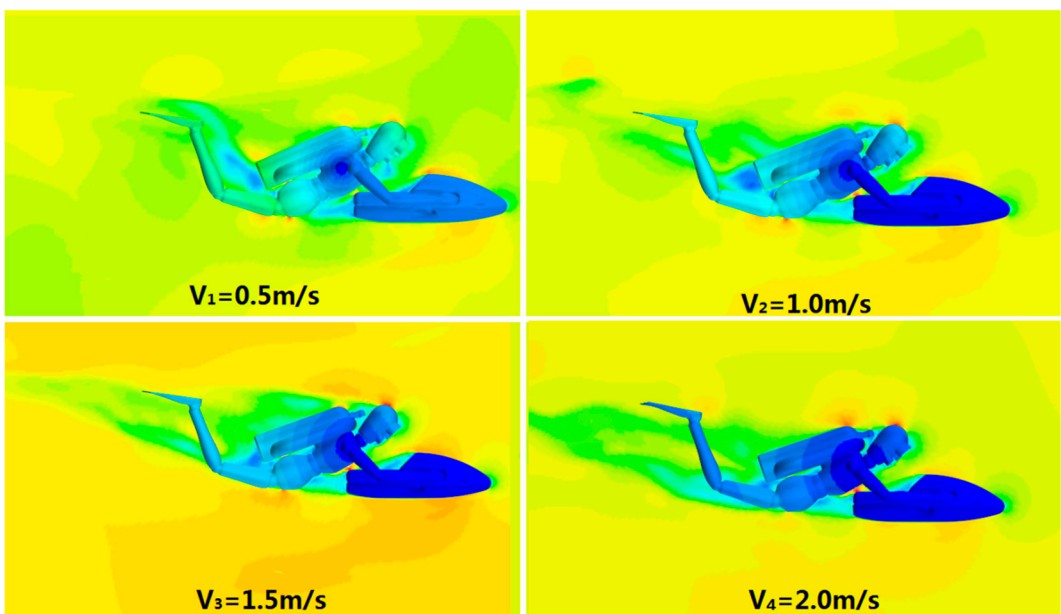

**Figure 21.** Velocity scalar nephogram at 2 s moment at other four cruising speeds.

The analysis shows that with the increase of speed *V*, the Body and Hip posture angle $\theta_{P1}$ decreases with the increase of speed, the Legs posture angle $\theta_{P4}$ is also the same, and the resistance $F_f$ gradually decreases. The fitting curve of the relationship between $F_f$ and $V^2$ is shown in Figure 22, and the fitting equation is in Table 7.

It can be seen that the resistance velocity relationship of traditional submersible is not consistent with $F_f = 1/2 C_t \rho_w v^2 S$, This is due to the change of human body posture angle under the action of flow field, which leads to the real-time change of resistance coefficient $C_t$, and finally tends to the minimum value. The specific process is as follows:

According to the Figure 20, the absolute value of lift $|F_l|$ under various working conditions has experienced a state from large to small, and finally tends to 0N. Therefore, in straight-line navigation, the lift force brought by the real-time posture angle $\theta_p$ (t) will drive the human body to approach the posture with the minimum resistance $F_l$ (t). When the final lift tends to 0, the resistance is the minimum, and the posture angle reaches the target value.

When the resistance $F_f$ converges, there is still a difference between the resistance of the constrained diver–DPV model, which is defined as the posture disturbance term $F_{dis}$. When $F_{dis} > 0$, it is called favorable disturbance, and when $F_{dis} < 0$, it is called adverse

disturbance. Its relationship with other resistance components and speed is discussed in the next section.

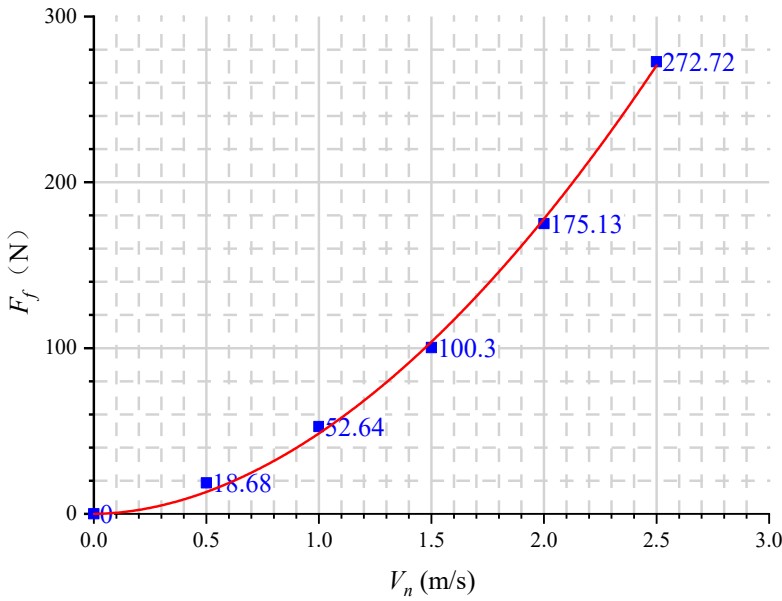

**Figure 22.** Fitting curve of the relationship between $F_f$ and $V_n$.

**Table 7.** Fitting curve equation of the relationship between $F_f$ and $V$.

| Equation | $F_f = a*v\hat{\ }b$ |
|---|---|
| $a$ | $48.51346 \pm 2.39272$ |
| $b$ | $1.87543 \pm 0.06026$ |
| Reduced chi-square | 18.02041 |
| $R^2$ (COD) | 0.99867 |
| Adjusted $R^2$ | 0.99834 |

## 4. Discussion

This section analyzes the relationship between the X-axis resistance disturbance term $F_{dis}$, speed and posture. Firstly, the total resistance components of the diver–DPV cruising in the water are analyzed. The results show that the total resistance of the diver–DPV coupled model in the water is $F_f = F_{f0} + F_{dis}$, where $F_{f0}$ is the resistance value of restricting the direct navigation movement. Ignoring the influence of near surface and bottom effect, the component of the resistance of $F_{f0}$ is usually divided into the following points: friction resistance $R_f$, residual resistance $R_{rs}$, Therefore, the total resistance can be expressed as

$$F_f = F_{f0} + F_{dis} = R_f + R_{rs} + F_{dis} \tag{16}$$

According to Hughes, the ratio of viscous resistance coefficient to frictional resistance coefficient is constant. By introducing the shape factor to deal with the three-dimensional flow of the diver–DPV model, the resistance conversion between the model and the real vehicle is realized. The resistance coefficient relationship is

$$C_t = C_{tm} - (1+k)\left(C_{fm} - C_{fs}\right) + \Delta C_f \tag{17}$$

where $C_{tm}$ is the resistance coefficient of the model, $(1 + k)$ is the shape coefficient, $C_{fm}$ is the friction resistance coefficient, $C_{fs}$ is the actual friction resistance coefficient, $\triangle C_f$ is the roughness conversion subsidy coefficient.

According to the formula of ITTC (International Towing Tank Conference) [25], the friction and resistance coefficient of a real vehicle can be calculated by Reynolds number (*Re*)

$$C_f = \frac{0.075}{(\lg Re - 2)^2} \tag{18}$$

$$Re = \frac{UL}{v} \tag{19}$$

where $v$ is the water viscosity coefficient affected by temperature.

The corresponding subsidy coefficient of roughness conversion can be expressed as

$$\Delta C_f = \left[ 105 \left( \frac{K_s}{L} \right)^{1/3} - 0.64 \right] \times 10^{-3} \tag{20}$$

$K_s$ is the height of rough surface. This CFD model is a full-scale model. The fresh water temperature is 25 °C and the kinematic viscosity coefficient of water is $v = 0.9167 \times 10^6$ m$^2$/s. Without considering the cruising wear, this paper takes $K_s = 150 \times 10^{-6}$. In conclusion, according to the main scale of diver–DPV model, the roughness conversion subsidy coefficient can be obtained, and then the resistance coefficient conversion between the CFD model and the actual DPV can be realized.

In order to effectively analyze the change of resistance components, the model resistance is dimensionless. Assuming that the wet surface area of diver–DPV model is constant, there is still friction resistance $f = 1/2C_f\rho_w v^2 S$. The empirical formula of friction resistance coefficient is

$$C_f = \frac{0.445}{(\lg Re)^{2.58}} \tag{21}$$

The total resistance coefficient is still calculated as $F_f = 1/2C_t\rho_w v^2 S$, while $F_{f0} = 1/2C_{t0}\rho_w v^2 S$.

That is:

$$C_t = C_{t0} + C_{dis} = C_f + C_{rs} + \triangle C_f + C_{dis} \tag{22}$$

According to that the wet surface area $S_w = 3.521381$ m$^2$, we get $C_{t0} = 27.2984/10^{-3}$. The results are shown in Table 8.

**Table 8.** Dimensionless results of diver–DPV resistance.

| Reynolds Number *Re* | Froude Number *Fr* | Total Resistance Coefficient $C_t/10^{-3}$ | Friction Resistance Coefficient $C_f/10^{-3}$ | Residual Resistance Coefficient $C_{rs}/10^{-3}$ | Disturbance Coefficient $C_{dis}/10^{-3}$ | Rough Subsidy Coefficient $\triangle C_f/10^{-3}$ |
|---|---|---|---|---|---|---|
| 1324.86 | 0.00867 | 42.4381 | 23.5864 | 18.1827 | 6.6641 | |
| 2649.72 | 0.03469 | 29.8993 | 18.6005 | 11.2988 | 4.5793 | |
| 3974.58 | 0.07944 | 25.3188 | 16.3426 | 8.97622 | −7.8424 | 0.669 |
| 5299.44 | 0.13875 | 24.8670 | 14.9653 | 9.90173 | −17.1235 | |
| 6624.30 | 0.21680 | 24.7833 | 14.0056 | 10.7777 | −27.6766 | |

According to Table 8, the resistance coefficient $C_t$ of the diver–DPV coupled model is plotted as Figure 23 according to the variation curve.

It can be seen from Figure 23 that the total resistance coefficient of model tends to decrease with the increase of Froude number $F_r$. The friction resistance $R_{fs}$ in those speed cases is the main part of the total resistance, and the correlation between the value and the speed change trend gradually decreases. When $F_r > 0.06$, $C_{dis} < 0$, which means it is favorable disturbance; when $F_r < 0.06$, $C_{dis} > 0$, which means it is adverse disturbance.

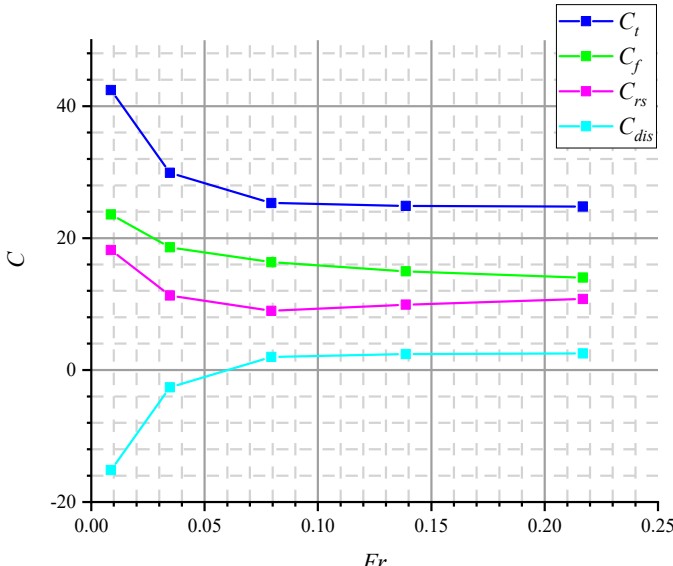

**Figure 23.** Variation of resistance coefficient of DPV underwater cruising.

According to various resistance coefficients, the actual resistance $F_f$, resistance of restraint movement $F_{f0}$, disturbance term $F_{dis}$ and speed $V$ are shown in Table 9.

**Table 9.** Resistance and speed.

| $V_n$ (n = 1~5) | $F_f$ (N) | $F_{dis}$ (N) | $F_{f0}$ (N) |
|---|---|---|---|
| 0.5 | 18.68 | 6.66 | 12.02 |
| 1.0 | 52.64 | 4.58 | 48.06 |
| 1.5 | 100.30 | −7.84 | 108.14 |
| 2.0 | 175.13 | −17.12 | 192.26 |
| 2.5 | 272.723 | −27.68 | 300.40 |

The fitting curves of $F_{dis}$ and $V_n$ is as follows in Figure 24:

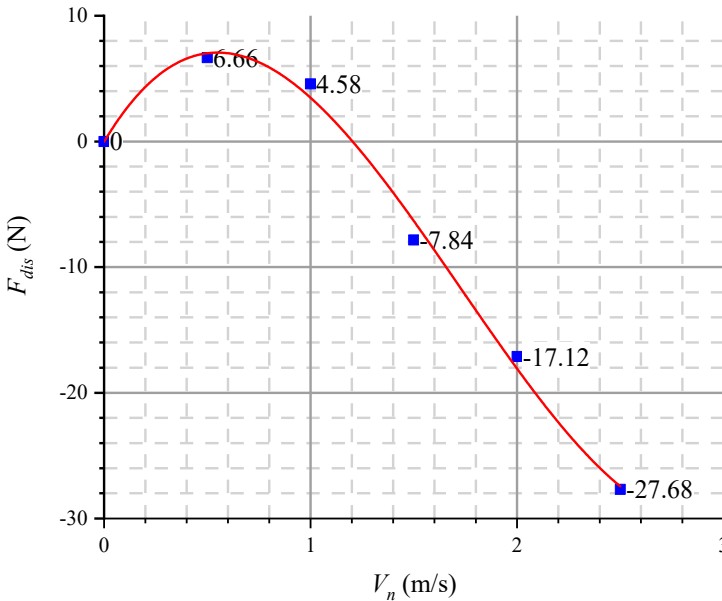

**Figure 24.** Fitting curve of the relationship between $F_{dis}$ and $V_n$.

Finally, the relationship between the total resistance $F_f$ and $V_n$ is obtained as

$$F_f = F_{f0} + F_{dis} = \tfrac{1}{2}C_{t0}\rho_w V_n^2 S + A + B*V_n + C*V_n^2 + D*V_n^3 \tag{23}$$

The diver–DPV disturbance in the *x*-axis is

$$m\dot{V}_n(t) = \tfrac{1}{2}C_{t0}\rho_w V_n(t)^2 S + A + B*V_n(t) + C*V_n(t)^2 + D*V_n(t)^3 \tag{24}$$

The values of the constants are shown in the Table 10.

**Table 10.** Fitting curve equation of the relationship between $F_{dis}$ and $V_n$.

| Equation | $F_{dis} = A + B*V_n + C*V_n^2 + D*V_n^3$ |
|:---:|:---:|
| *A* | $-0.0299 \pm 1.48536$ |
| *B* | $27.55986 \pm 5.77036$ |
| *C* | $-29.82083 \pm 5.73532$ |
| *D* | $5.76419 \pm 1.50635$ |
| Reduced Chi-Sqr | 2.2975 |
| $R^2$(COD) | 0.9949 |
| Adjustment $R^2$ | 0.9872 |

From the conclusion, it can be concluded that in the design of DPV, if the maximum cruising speed index is too low, the diver's attitude will always be in the range of adverse disturbance, and the navigation efficiency will be reduced. However, if the maximum cruising speed is too high, the disturbance coefficient will not change, and the cruising efficiency will not be further improved. It can be understood that the high-speed flow field has already impacted the diver's body to the horizontal attitude, and will not change further. Through this CFD model, we can get the most suitable speed range index in the DPV design, which is of great significance to the design of various shape of DPV in the future.

## 5. Conclusions

In summary, by analyzing the existing DPV (diver propulsion vehicle) equipment, a set of diver–DPV multi-body coupling model considering rigid body dynamics and fluid disturbance is established in this paper. The split and motion of human joints is realized by overlapping grid of Star-CCM+ software and DFBI 6-DOF model motion method. The nonlinear disturbance caused by diver posture transformation under 10 different cruising cases and its influence on the rapidity and diver posture characteristics of DPV are concluded as follows:

When a diver driving DPV and cruising straight line, the angle of diver body's posture is related to the speed of the vehicle. The movement of this posture change produces a disturbing potential to the surrounding flow field, and produces a resistance disturbance term to the diver–DPV coupled model, which makes the overall cruising resistance different from the resistance at the same speed under constraint human body conditions. The relationship between the sign and volume of the disturbance term $F_{dis}$ and the speed is as follows: $F_{dis} = A + B*V_n + C*V_n^2 + D*V_n^3$, when $Fr > 0.06$, $F_{dis} < 0$, which means it is favorable disturbance and can reduce cruising resistance. When $Fr < 0.06$, $F_{dis} > 0$, which means it is adverse disturbance. In addition, within the design maximum speed of DPV, the friction resistance $F_f$ is always the main part.

Through this multi-body hydrodynamic model, we can get the most suitable speed range index in the DPV design. It is of great significance for the future design of DPV with various shapes to improve the underwater cruising efficiency.

**Author Contributions:** Conceptualization, H.L. and H.Z.; methodology, F.H.; software, W.Z.; valida­tion, H.L and Y.W.; formal analysis, J.Z.; investigation, H.L and H.Z.; data curation, J.Z.; writing—original draft preparation, H.L; writing—review and editing, W.Z. and F.H.; visualization, W.Z.; supervision, H.Z. All authors have read and agreed to the published version of the manuscript.

**Funding:** This research was funded by National Key R&D Program of China, grant number 2018YFC0309402.

**Data Availability Statement:** Not applicable.

**Acknowledgments:** First of all, I would like to appreciate for Harbin DepTech marine technology Co., Ltd. and other corporations for the DPV product data. Second, thanks to every scuba diving instructors from PADI for their detailed introduction on the professional DPV driving posture. Finally, I would like to thank all the editors and reviewers for their careful guidance and help. This is the first time that I have written an international top-level journal paper as the first author. Thanks a lot for your tolerance of minor errors in the process of paper revision. Wish there will be more cooperation in the future.

**Conflicts of Interest:** The authors declare no conflict of interest.

## Abbreviations

| | |
|---|---|
| $Re_{min}$ | Reynolds number of minimum rigid body in multi joint model |
| $Fr$ | Froude number, $Fr = v^2/gL$ |
| $V$ | Dynamic viscosity coefficient of fresh water |
| $L_{DPV}$ | Length of DPV |
| $B_{DPV}$ | Width of DPV |
| $H_{DPV}$ | Height of DPV |
| $D_{max}$ | Maximum diving depth |
| $m$ | Weight of DPV |
| $L$ | Length of the diver–DPV coupled model |
| $B$ | Width of the diver–DPV coupled model |
| $H$ | Height of the diver–DPV coupled model |
| $M$ | Weight of the diver–DPV coupled model |
| $a_x$, $a_y$ | Acceleration of $x$-axis and $y$-axis cruising |
| $P_n$ | Hinge points of multi rigid body joint model, n = 1, 2, 3, 4, 5 |
| $\theta_{Pn}$ | Angle between diver joint and $x$-axis at hinge point n |
| $C_t$ | Total resistance coefficient |
| $C_{dis}$ | Disturbance resistance coefficient |
| $C_f$ | Friction resistance coefficient |
| $C_{rs}$ | Residual resistance coefficient |
| $\triangle C_f$ | Rough subsidy coefficient |
| $C_{dis}$ | Disturbance coefficient |
| $V_n$ | Underwater cruising speed of DPV, n = 1, 2, 3, 4, 5 |
| $S_w$ | Wet surface area of the model |
| $I_x$, $I_y$, $I_z$ | Moment of inertia for $X$, $Y$, $Z$ axis |
| $F_t$ | Propeller thrust |
| $F_f$ | Cruising resistance of the diver–DPV model |
| $F_{f,Vn}$ | Cruising resistance at different speeds |
| $F_{f,meshn}$ | Resistance at maximum speed $V_5$ with different computational grid, n = 1, 2, 3 |
| $F_b$ | Buoyancy |
| $F_l$ | Lift force |
| $F_{l,da}$ | Z-axis lift force acting on DPV and arms |
| $F_{f,da}$ | Resistance in $x$-axis direction acting on DPV and arms |
| $F_{b,da}$ | Buoyancy of DPV and arms |
| $G_{da}$ | Gravity of DPV and arms |
| $F_{P2X}$, $F_{P2Y}$ | Tensile force at hinge point $P_2$ |
| $F_{l,bh}$ | Lift force acting on body and hip in $z$-axis direction |
| $F_{f,bh}$ | Resistance in $x$-axis direction acting on body and hip |
| $F_{b,bh}$ | Buoyancy of body and hip |

| | |
|---|---|
| $G_{bh}$ | Gravity of body and hip |
| $F_{P4Y}, F_{P4X}$ | Tensile force at hinge point $P_4$ |
| $F_{l,leg}$ | Z-axis lift acting on legs |
| $F_{f,legs}$ | Resistance in *x*-axis direction acting on legs |
| $F_{b,legs}$ | Buoyancy of legs |
| $G_{legs}$ | Gravity of legs |
| $F_{dis}$ | Disturbance force |
| $F_{disn,x}$ | Disturbance force on different 6-DOF body in *x*-axis, n = 1, 2, 3 |

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
