# Peer review of "Hydrodynamic Model of Diver–DPV Coupled Multi-Body and Its Underwater Cruising Numerical Simulation"

_jmse, doi:10.3390/jmse9020140_

Round 1

Reviewer 1 Report

The paper makes a case in favor of using the SST turbulence model, but the way the CFD study was conducted is inconsistent with the prescribed SST guidelines. Instead the y+ distribution (above 10) would indicate that the actual model used by the authors is the standard k-epsilon (which is what the SST uses in the far field). This is not proper, because in the boundary layer modeling, SST should use the k-omega model, which is more accurate. 

Hence either a new case in favor of the k-epsilon model is provided, showing how it is still a valid way to carry out CFD in these cases, or the simulations must be re-done with a finer mesh, consistent with the SST model.

Author Response

Dear reviewer,

My appreciation  for the detailed comments. All the responses are attached in the file. If there's any problem, please be no hesitated tu contact with us.

Yours sincerely,

The authors.

Reviewer 2 Report

Dear authors, 

Please find my comments and suggestions attached below. Thanks!

Author Response

(The authors gave the same response as above.)

Round 2

Reviewer 1 Report

unfortunately my personal feeling leans towards remaking the simulations, since the y+ is one order of magnitude higher than the recommended one for the SST turbulence model which is employed.

However the final decision rests with the editor and if this is considered a minor problem, then I will not further request any modifications.

Reviewer 2 Report

Dear Authors,

Thank you for clearly addressing all of my comments in regard in to your manuscript. I greatly enjoyed reading through it and believe it is a fascinating study and I look forward to seeing your future work on the topic.

This manuscript is a resubmission of an earlier submission. The following is a list of the peer review reports and author responses from that submission.

Round 1

Reviewer 1 Report

The paper addresses a very interesting topic and is on the right track in terms of methodology. That having been said, there are certain problems that need to be fixed before it can be considered for publication, please see below:

Abstract is marginally well written, it may emphasize more on the reader's interest, but it can be accepted as it is.

The introduction is well done, but only 4/21 references are newer than 5 years, which begs the question on how the research integrates in the state of the art.

Make sure that the referencing is properly done, i.e. in the correct format.

Make sure that all figures and tables are called out in the text and have meaningful captions in the context of your argumentation

The flow equations are well known, there is no need to have them in the paper

There are many notations described inside the paper body, please consider making a nomenclature section instead.

If you are going to discuss convergence and case stability, please include CFL number, numerical scheme and convergence criteria, not just the sub-relaxation factors and number of total iterations

The case y+ distribution is missing, which means we cannot trust that the k-omega SST model was correctly implemented. Also the grid dependency is addressed in a sketchy way, which also adds to the uncertainty of the cases.

Perhaps the most interesting conclusion is that of the favorable disturbance which is mentioned only in passing in the Conclusion section. I recommend an expansion of this and a truncation of the conclusions which were either to be expected or and have no quantitative data attached to it.